# Angiogenesis and Multiple Sclerosis Pathogenesis: A Glance at New Pharmaceutical Approaches

**DOI:** 10.3390/jcm11164643

**Published:** 2022-08-09

**Authors:** Maria Teresa Gentile, Gianluca Muto, Giacomo Lus, Karl-Olof Lövblad, Åsa Fex Svenningsen, Luca Colucci-D’Amato

**Affiliations:** 1Laboratory of Cellular and Molecular Neuropathology, Department of Environmental, Biological and Pharmaceutical Science and Technology, University of Campania “L. Vanvitelli”, 81100 Caserta, Italy; 2Division of Diagnostic and Interventional Neuroradiology, Geneva University Hospitals, 1205 Geneva, Switzerland; 3Multiple Sclerosis Center, II Division of Neurology, Department of Advanced Medical and Surgical Sciences, University of Campania “L. Vanvitelli”, 81100 Caserta, Italy; 4Department of Neurobiology, Institute of Molecular Medicine, University of Southern Denmark, 5000 Odense, Denmark; 5InterUniversity Center for Research in Neurosciences (CIRN), University of Campania “Luigi Vanvitelli”, 80131 Naples, Italy

**Keywords:** multiple sclerosis, angiogenesis, neuroprotection

## Abstract

Multiple sclerosis is a chronic disease of the central nervous system characterized by demyelination and destruction of axons. The most common form of the disease is the relapsing-remitting multiple sclerosis in which episodic attacks with typical neurological symptoms are followed by episodes of partial or complete recovery. One of the underestimated factors that contribute to the pathogenesis of multiple sclerosis is excessive angiogenesis. Here, we review the role of angiogenesis in the onset and in the development of the disease, the molecular mechanisms underlying angiogenesis, the current therapeutic approaches, and the potential therapeutic strategies with a look at natural compounds as multi-target drugs with both neuroprotective and anti-angiogenic properties.

## 1. Introduction

Multiple sclerosis (MS) is a chronic disease of the central nervous system (CNS) that affects approximately 2.3 million people worldwide [1] and is currently incurable. Its debilitating symptoms range from numbness and tingling to more severe partial or complete paralysis. MS is typically diagnosed between ages 20 and 40 [1,2], can substantially and adversely affect an individual’s quality of life, and is associated with high costs for society. MS is mostly described as a relapsing-remitting form (RRMS) that is characterized by episodic attacks, with typical neurologic symptoms (relapses or exacerbations), followed by episodes of partial or complete recovery (remission) [3]. However, some patients can acquire disability also through a relapsing independent progression, a condition known as progression independent of relapse activity (PIRA). In particular, PIRA has been shown to occur in typical RRMS populations, challenging the traditional distinction between an early exclusively relapsing phase and a late secondary progressive MS (SPMS). The pathophysiological determinants of PIRA are not understood, although recent studies show that PIRA is associated with increased diffuse neuroaxonal loss, and accelerated brain atrophy, especially in the brain cortex [4]. From the pathogenic point of view, MS is characterized by loss of myelin and destruction of axons, infiltration and activation of inflammatory cells, and blood–brain barrier (BBB) breakdown [5,6]. It is unknown whether MS has a single or a multiple cause, and several hypotheses have been postulated on the first cause of the disease. It is likely that it is due to genetic predisposition and environmental factors, such as infectious, vascular, or metabolic factors, that combine to initiate the disease [3]. On one hand, epidemiology of MS indicates that the risk of developing the disease increases to 2–4% in people with first degree affected relative, and to 30–50% in monozygotic twins. Indeed, genome-wide association studies proved that the haplotype HLA DRB1*1501 is the most significant gene variant that raises the risk of the disease. On the other hand, several environmental risk factors have been identified. It has been reported that, in geographic latitudes with more temperate climates, there is a higher incidence of the disease. This could be due to seasonal exposure to sunlight that influences the synthesis of vitamin D. Other environmental risk factors associated with an enhanced possibility to develop the disease are cigarette smoking, obesity, and mononucleosis [3]. Thus, MS is not simply a single disease entity but could derive from multiple pathogenic factors that lead to final events represented by an inappropriate recognition of an autoantigen on myelinated nerve fibers that recruit macrophages and lymphocytes into the CNS. Another view regards MS as a primary neurodegenerative disease modified and exacerbated by the following inflammation [7]. Other telltale signs of MS are gliosis, axon degeneration, and spontaneous attempts at remyelination. Moreover, studies suggest that, in MS demyelinating lesions, following vascular endothelial growth factor (VEGF) release, new blood vessels are formed [8].

## 2. Angiogenesis in Multiple Sclerosis

The complex process that leads to the formation of new blood vessels from preexisting ones is called angiogenesis. In this process, endothelial cells proliferate, differentiate and migrate to form an intact vascular network. During embryogenesis and after birth, angiogenesis contributes to organ growth but, in adulthood, most blood vessels remain quiescent and angiogenesis occurs only in the ovarian cycle, in the placenta during pregnancy and during wound healing and repair [9]. Within the CNS, the process of angiogenesis is integrated with a series of programmed changes in endothelial cells, which end in the formation of a tight barrier called the blood–brain barrier (BBB). The term blood–brain barrier describes the microvasculature of the CNS whose function depends on the ‘neurovascular unit’ composed of neurons, astrocytes, and microglia, and, more importantly, endothelial cells of the capillaries. Endothelial cells of cerebral vessels are devoid of fenestrations and trans-endothelial channels. For these reasons, the BBB is a highly selective semipermeable membrane barrier that separates the circulating blood from the brain and extracellular fluid in the CNS. It allows the passage of water, some gases, and lipid-soluble molecules by passive diffusion, as well as the selective transport of molecules such as glucose and amino acids that are crucial for neural function. Thus, for the most part, the BBB prevents the movement of substances from the blood to brain tissues [10,11]. In MS, BBB function fails because of a dysregulated paracellular permeability due to the release of histamine, leukotrienes, and chemokines, which recruits neutrophils and other cells to invade CNS [12]. In particular, in MS, the permeability of the BBB is altered with subsequent transmigration of different subtypes of lymphocytes (Th1 and Th17) and inflammatory cells such as macrophages into the CNS and production of inflammatory mediators (e.g., TNF-α and IFN-γ). This represents an early event in MS pathogenesis and precedes other clinical neurological signs [13]. Moreover, it is well known that chronic inflammation and angiogenesis are mutually related and that the immune cells, in particular mast cells, involved in MS pathogenesis are able to secrete cytokines that activate endothelial cells inducing their proliferation and migration [14]. Imbalance in angiogenic processes, including excessive angiogenesis, is a hallmark of a number of diseases such as malignant ocular inflammatory disorders, obesity, diabetes, cirrhosis, endometriosis, AIDS, and rheumatic diseases [9]. In these disorders, the balance between angiogenesis stimulators and inhibitors is impaired, resulting in an “angiogenic switch” [15]. Several studies indicate a role of angiogenesis in MS in humans [9,16,17] as well as in vitro using sera of MS patients in a chick embryo CAM assay [18]. Moreover, magnetic resonance imaging (MRI) studies on cerebral perfusion in MS patients show significant increases in local blood flow and blood volume related to disease progression. These results indicate an increase in blood vessel density and vasodilation. In particular, Bester et al. reported a study in which 33 patients with relapsing-remitting MS underwent four monthly 3T MRI scans including dynamic susceptibility contrast perfusion-weighted MRI. They show that high inflammatory lesions are associated with increased cerebral blood volume (CBV) and cerebral blood flow (CBF) [19]. CBV has been proven to be a sensitive marker of angiogenesis, with significant correlations between the increase of CBV and CBF values and microvessel density [20]. Notably, the arrangement of angiogenic vessels in MS appears orderly, whereas tumor angiogenesis is chaotic [21], suggesting a reparative response to injury [22]. Although no immunohistochemical data are currently available, it is possible that the elevated CBV/CBF values in the normal appearing white matter may be indicative of angiogenesis occurring additionally to inflammation-induced vasodilatation [23,24]. Moreover, clinical findings demonstrate that patients treated with natalizumab (NTZ), a monoclonal anti-α4β1 integrin (VLA-4) antibody which prevents the lymphocytes transmigration across the BBB, show a significant reduction of local blood flow [25,26,27], suggesting that natalizumab may exert its anti-inflammatory properties partly by inhibiting the angiogenic mechanisms in relapsing-remitting MS patients [28,29]. Two hypotheses have been postulated about the role of angiogenesis in MS. In the early phase of the disease, angiogenesis may allow the recruitment of peripheral immune response cells. Blood borne macrophages infiltrate active multiple sclerosis lesions and remove myelin debris and inflammatory by-products; classically and alternatively activated macrophages, as well as mixed populations, have been described in these lesions [3]. On the other hand, in the later phases of MS, angiogenesis could represent an attempt to rescue the hypo-perfusion due to the reduction of vascular permeation in CNS [14]. In order to find sensitive and precise peripheral biomarkers of MS, useful for making an early diagnosis, recent research has focused attention on exosomes and small non coding RNA molecules. It is well known that extracellular miRNA performs a wide spectrum of functions, such as transmitting signals between cells, modulating processes involved in angiogenesis, neurogenesis, proliferation or apoptosis [30], myelination [31], and inflammatory response [32]. In particular, studies on patients and EAE models identified miR155 as a key regulator of inflammatory response involved in MS, such as Th1 and Th17 differentiation, both in CNS and peripheral blood. Moreover, miR-301a was found to be a critical regulator of myelin-reactive T-h 17 cells, further supporting their role in controlling autoimmune demyelination [32]. Other studies evidenced that miR-23a enhances both oligodendrocyte differentiation and myelin synthesis since miR-23a-overexpressing mice have increased myelin thickness [31]. Indeed, in a recent study, a large group of MS patients (*n* = 1088) was stratified based on brain imaging, and miRNA profiling was used to identify differences across the different magnetic resonance imaging (MRI)-based phenotypes [33]. Interestingly, each MRI phenotype demonstrated the presence of a specific miRNA whose function is involved in blood–brain barrier (BBB) pathology. Specifically, miR-22–3p, miR361–5p, and miR-345–5p, all involved in the angiogenesis process [34,35,36], were the most valid differentiators of the MRI phenotypes [37]. However, it must be underlined that, on this issue, there are conflicting results, heterogeneity of data, and lack of replication and that there is still a lot of work in progress regarding the mechanism of miRNA networks and their targets.

## 3. VEGF: A Double Edge Sword in the Treatment of MS

Experimental Autoimmune Encephalomyelitis (EAE) represents the most commonly used animal model for MS. It is induced by injecting mice with antigens derived from myelin. These antigens elicit an acute activation of T-cells and macrophages with a subsequent demyelinating process which can have a chronic relapsing course quite similar to MS [38]. It has been reported that EAE induced in C57Bl/6 mice immunized by Myelin Oligodendrocyte Glycoprotein (MOG35–55) shows an early BBB disruption in the CNS and an increased vessel density and length in the cerebral cortex. In MOG-induced EAE, vascular remodeling is also an early process where increased vessel area and endothelium proliferation appear as early as four days after immunization, in a pre-symptomatic disease phase [39,40]. Several reports suggest that vascular endothelial growth factor (VEGF) is involved in the pathogenesis of MS. An increased VEGF expression has been observed in reactive astrocytes of both active and inactive chronic demyelinated lesions, in normal-appearing white matter from post mortem MS brains, in the sera of MS patients during clinical disease relapses and is correlated with the length of spinal cord MS lesions [41,42,43,44,45]. It has also been shown that the inhibition of VEGF during EAE progression improves the clinical score and attenuates both demyelination and inflammation [46,47]. VEGF within the CNS can also modulate the immune cell response. In fact, VEGF and its receptor 2 (VEGFR2) are present both as mRNA and protein in T-lymphocytes in MS hypoxic lesions. VEGF exposure also induces a shift to a Th1 pro-inflammatory profile of those T lymphocytes, with increased interferon-gamma secretion and decreased interleukin 2 productions [48,49]. To analyze the effects of compounds on the effectors’ phase of EAE, independently of the immunization process, it is recommended to use the so-called adoptive transfer EAE model. In this model, T-cells from an immunized donor mouse are transferred into recipient mice, where they induce EAE (effectors’ phase of EAE). The model is also used to test the effects of compounds on differentiated Th1 or Th17 lymphocyte populations as well as cell trafficking [50]. When adoptive EAE transfer was induced by an injection of T-lymphocytes pretreated with VEGF, the disease started earlier and became more severe. In fact, activated T-cells can secrete VEGF and promote angiogenesis, which, in turn, may act in an autocrine fashion to exacerbate the inflammatory response [51]. Moreover, when bevacizumab, a monoclonal anti-VEGF antibody that blocks the VEGF receptor, was administered intravenous to EAE mice on the day they began to show clinical symptoms, a significant reduction of the number of CD4+ T-cells and inhibition of T-cells proliferation in the CNS was observed [45]. However, intravenous administration of bevacizumab is also associated with a significantly increased risk of cerebrovascular events, including cerebral hemorrhages and ischemia [52]. It is worth noting that VEGF also exerts a well-known neuroprotective function, enhancing axonal growth and neural resistance to injury [53]. In addition, VEGF inhibits the synthesis of a wide spectrum of inflammatory cytokines in microglia/macrophages. This anti-inflammatory effect is linked to the plasticity-promoting action of VEGF [15]. VEGF receptors are upregulated on microglia and other antigen-presenting cells after CNS trauma, suggesting a modulating role of VEGF in CNS immune surveillance [54]. Despite its multifaceted role, the main target of antiangiogenic therapeutic strategies is in fact VEGF, which plays a predominant role in angiogenesis. Several antiangiogenic drugs also act by targeting the VEGF signaling pathways and some of them are now being tested in clinical trials. In particular, bevacizumab is being tested in patients with neuromyelitis optica, an aggressive inflammatory disease of the central nervous system, characterized by recurrent episodes of optic neuritis and transverse myelitis, which has many symptoms in common with MS [55,56,57]. However, a significant number of the patients in this group remain unresponsive and these synthetic drugs have an unfavorable cost-to-benefit ratio [57]. Moreover, it has been reported that, in the EAE murine model, antagonizing the VEGFR2 with one of these monoclonal anti-VEGF antibodies, smaxinib, was effective only in the acute inflammatory phase of the disease, but not in the chronic, neurodegenerative phase [56].

## 4. MEK/ERK1/2 Pathway: Essential to Angiogenesis Is Also a Hub Connecting Different Pathogenic Factors of MS

From the molecular standpoint, angiogenesis requires coordinated regulation by multiple signaling pathways, in particular, several growth factors, including VEGF and activated MAPK (mitogen-activated protein kinases), that play a pivotal role in cell survival, proliferation, and differentiation of endothelial cells during angiogenesis [57,58]. Several studies suggested that ERK1/2 (extracellular signal-regulated kinases; ERK) is also part of important processes critical for angiogenesis such as cytoskeleton structure, cell morphology and motility in endothelial cells [59,60], network formation, and increase in vessels lumen size [61]. Indeed, being activated by numerous membrane receptors after the interaction with their ligand, the ERK pathway is a convergent signaling node receiving input from numerous stimuli, including intracellular metabolic pathways as well as signaling from extracellular growth factors, cell–matrix interactions, and communication from other cells [61,62,63,64]. Thus, aberrant ERK signaling is involved in a variety of different pathological conditions including carcinogenesis [65], autoimmunity, and neurodegeneration [66,67,68,69]. ERK signaling is downstream of several neurotrophic factors and ERK activation is involved in neural survival and oxidative stress in vitro [70,71]. In vivo, animals with constitutively active ERK show significant hypertrophy of neuronal somata as compared to wild-type animals, and ERK phosphorylation is neuroprotective in these mutants [72]. ERK is also central in the regulation of immune cell function and has been implicated in the onset of autoimmune disorders. Recently, Birkner et al. have inhibited nuclear translocation of ERK in lymphocytes obtained from the EAE mouse model, demonstrating a significant reduction in vitro of the GM-CSF from Th17 that promotes the encephalitogenic of the T-lymphocytes [73]. In vivo studies, using the EAE model, have shown that pharmacological inhibition of the MEK-ERK pathway ameliorated the symptoms [74,75,76]. Recently, it has been showed that over-activity of ERK in microglial cells is linked to oligodendrocyte demyelination, highlighting that well-known risk factors for MS, such as hypo-vitaminosis D, tobacco smoking as well as Epstein–Barr Virus infection, are known to downregulate ERK by means of dual specificity phosphatases that normally regulate ERK phosphorylation and activity [76]. In conclusion, the MEK/ERK pathway, being involved in many aspects of MS pathogenesis, is a potential hub in the future development of novel therapeutic strategies against multiple sclerosis (Figure 1).

## 5. Therapy of Multiple Sclerosis

In recent years, the therapeutic approach to multiple sclerosis has been completely revolutionized thanks to the better clarifications of the pathogenic mechanisms. Almost 30 years after the first specific drug for the therapy of MS (beta interferons), numerous drugs are currently available, each with their own characteristics but all with a need for continuous treatment to make inflammation control lasting; some of these (including alemtuzumab and cladribine) cause an immune reconstitution and are administered for short periods in the hypothesis of producing lasting immunological actions [74,75,76]. Despite this, all these therapies seem to act indirectly on neurodegeneration through the control of neuro-inflammation. However, the mechanisms of action of some available therapies and the pathogenesis of MS are still incompletely known, thus specific treatments to reduce neural damage or repair are not yet available. To date, angiogenesis is not considered the crucial event in the pathogenesis of MS. However, the role of angiogenesis in the different phases of disease progression makes this event an important and underestimated target in the development of new therapeutic strategies. The inhibition of angiogenesis in the early phase of MS could potentially prevent the vascular supply of nutrients and inflammatory cells to the demyelinating lesions, limiting the production of endothelial-derived pro-inflammatory cytokines. Future therapeutic efforts should consider the natural progression of the disease in which angiogenesis is involved in both inflammation, in the remitting phase, as well as in neurodegeneration in the progressive phase.

## 6. Natural Compound in the Therapy of Multiple Sclerosis

It is imperative to develop new antiangiogenic drugs with minimal side effects on the neural component to complement and combine with existing MS therapies. It would be possible to find such agents in plants where some natural compounds are characterized by high chemical diversity and biochemical specificity. Many active substances from medicinal plants, for example, flavonoids, polyphenols, furocoumarins, and related molecules have been tested for their potential therapeutic effects in diseases such as cancer, inflammatory and cardiovascular diseases [77]. A lot of studies report the anti-inflammatory and antioxidant effects of medicinal plants, as well as other helpful properties which make them a natural, often safe, and reliable remedy for the treatment of neurodegenerative diseases [78,79]. The use of complementary and alternative medicine (CAM), in particular, herbal remedies, has noticeably risen in MS patients over the last few decades [80,81,82,83]. The beneficial effects of CAM in MS disease occur by reduction of the severity of MS disease, including antioxidative properties, anti-apoptotic effects, anti-inflammatory properties, and promoting the differentiation of local stem cells to myelin-producing cells [84]. Different herbal medicines showed positive effects for MS patients. For a comprehensive review of this topic, refer to Mojaverrostami et al., who discuss some of the herbal compounds’ beneficial effects on the MS disease [84]. It is worth mentioning that the water extract of *Ruta graveolens* (RGWE) has antiangiogenic properties [85,86,87] as well as neuroprotective ones [86,87]. This compound can inhibit ERK1/2 signaling in endothelial cells (HUVEC) and significantly reduces VEGF expression [85,88]. The analysis of the biochemical composition of the extract revealed molecules such as epigallocatechin, a compound known to inhibit VEGFR expression [88], isoquercitrin, that inhibits the Wnt/β-catenin pathway [89] and rutin with potent anti-oxidant properties [89]. Thus, RGWE represents a compound in which polyphenolic molecules, already known to act as antiangiogenic compounds at higher concentrations, are perfectly combined to exert synergistic effects. Other studies have previously shown that the alcoholic extracts of *R. graveolens* prevent inflammation and angiogenesis associated with atherosclerosis in hypercholesterolemic rats [90]. These results suggest that RGWE could be a potential therapeutic tool able to inhibit undesired angiogenesis and to prevent inflammation as well as neurodegeneration. Other natural compounds have been tested for their potential therapeutic effects but none of them has been tested for both antiangiogenic and neuroprotective properties. Ip et al. have reported that anemoside, a natural triterpenoid glycoside isolated from the root of *Pulsatilla chinensis*, acting as a non-competitive NMDA receptor modulator, has neuroprotective effects and, concomitantly, an anti-inflammatory effect, thus resulting in being very promising in the therapy of neuroinflammatory disorders such as MS [91]. *Cannabis sativa* represents the most common plant used in the therapy of multiple sclerosis. It is well recognized that it has anti-inflammatory and immunomodulatory properties, as well as analgesic effects [92]. The use of *C. sativa* by MS patients reduced muscle stiffness, bladder disturbance, spasms, neuropathic pain, and sleep disorders [93]. Other plants known to have neuroprotective and anti-inflammatory effects are *Ginkgo biloba*, *Zingiber officinalis, Curcuma longa*, *Oenothera biennis,* Nigella *sativa, Crocus sativus, Panax ginseng, Boswellia papyrifera, Vitis vinifera, Gastrodia elata,* and *Camellia sinensis* [94,95,96]. It is worth noting that chronic inflammation and neo-angiogenesis can be strictly linked since a number of immune cells such as macrophage, neutrophils, dendritic cells, lymphocytes, and mast cells secrete growth factors and cytokines capable of fostering angiogenesis from endothelial cells. In particular, a role of mast cells has been recently proposed in fostering angiogenesis in MS. According to experimental findings, mast cells promote angiogenesis both directly by secreting a number of angiogenic factors (i.e., VEGF, FGF-2 etc.) and indirectly by stimulating other inflammatory cells to produce angiogenic molecules [14]. Although further studies are needed to disclose the exact mechanisms of action, through which medicinal plants exhibit their anti-inflammatory and neuroprotective effects, due to the complex nature of the pathogenesis of MS, multifunctional drugs with both neuroprotective and anti-inflammatory and, possibly, antiangiogenic properties are appealing for research in MS drug discovery. Moreover, as already occurred in other instances, “omic” approaches, such as genomic, proteomic, and epigenetic techniques, associated with high-throughput screening and machine learning, might shed lights into intracellular molecular targets as well as selective compounds within medicinal plants [97,98,99].

## 7. Conclusions

The history of medicine teaches us that many drugs currently used derive directly or indirectly from higher plants and, despite the ability to synthesize highly selective drugs able to target a single molecule rather than the organ or the entire organism, natural product research can still give important contributions to drug innovation by providing novel chemical structures and/or mechanisms of action. In the case of multiple sclerosis, the need to develop a multi-target drug able to act simultaneously on different pathogenic factors involved is imperative and could better improve both patient’s quality of life and the outcome of the disease. For example, the link between inflammation and angiogenesis, both involved in the pathogenesis of MS, makes compounds endowed with anti-inflammatory as well as anti-angiogenic properties particularly useful. Mounting evidence suggests that natural derivatives could be such compounds, able to exert an efficacious therapeutic effect and some of them are already used in clinical trials. Thus, naturally derived drugs are promising and affordable tools with limited side effects that can represent the future of therapeutic approaches to multifactorial diseases such as MS.

## Figures and Tables

**Figure 1 jcm-11-04643-f001:**
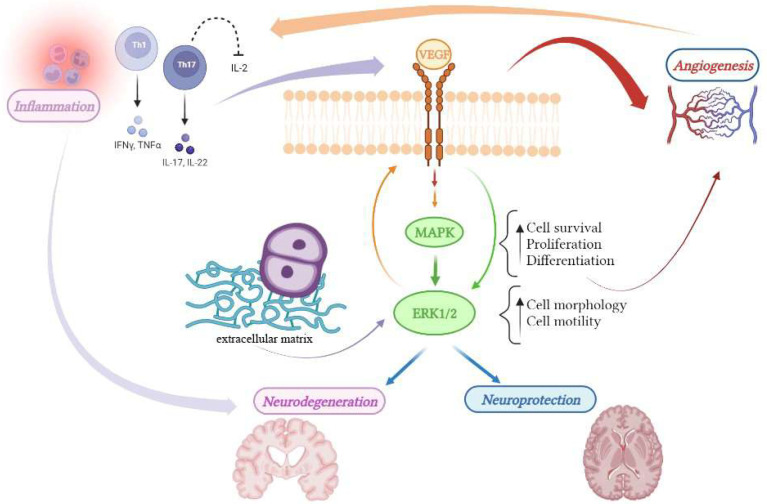
The MEK/ERK pathway represents the hub that connects many pathogenic factors involved in the onset of multiple sclerosis: inflammation, angiogenesis, and neurodegeneration. VEGF, via interaction with its receptor VEGFR, is the key protein that activates angiogenesis in response to inflammatory stimuli and is known to be neuroprotective. VEGF activates the downstream MEK/ERK pathway and is the main target of the current anti-angiogenic therapies. Moreover, ERK can be also activated in other ways, for example in microglial cells, thus contributing to inflammation and oligodendrocytes’ alterations.

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
