# Peer review of "Angiogenesis and Multiple Sclerosis Pathogenesis: A Glance at New Pharmaceutical Approaches"

_jcm, 2022, doi:10.3390/jcm11164643_

Round 1

Reviewer 1 Report

Dear Authors,

I read with great interest your paper entitled “ANGIOGENESIS AND MULTIPLE SCLEROSIS PATHOGENESIS: a glance at new pharmaceutical approaches”, the topic is interesting although the paper has serious flaws.

First, introduction does not provide a soundness background about the topic. Among the Cap 2 called “Angiogenesis in MS”, the role of angiogenesis in MS is not fully discussed. Moreover, the utility of Fig1. as a demonstration of angiogenesis in MS remains unclear and irrelevant (you described a typical MRI findings in a MS patient). Cap 3 and 4 are the only that contain relevant information about the topic thereby these paragraphs need to be better developed. In the cap 5, the description of this DMT is not pertinent with the topic “angiogenesis” , moreover table 1 is unnecessary in your manuscript (why you should report data about the administration and side effects of fingolimod, natalizumab, ocrelizumab and others?) , moreover it also contains incorrect data (fingolimod and natalizumab belong to immunomodulators). Conclusions are not pertinent with the aims of your paper. Finally, the entire manuscript requires an extensive language revision.

Author Response

Dear Referee,

we have amended the ms according to your comments.

We wish to thank you for the insightful criticisms and comments that helped us to amend and to ameliorate the ms.

Reviewer 1

First, introduction does not provide a soundness background about the topic.

We agreed that Introduction was poor concerning the background. Thus, we added the following points:

However, some patients can acquire disability also through a relapsing independent progression (PIRA). In particular, PIRA has been shown to occur in typical RRMS populations, challenging the traditional distinction between an early exclusively relapsing phase and a late secondary progressive MS (SPMS). The pathophysiological determinants of PIRA are not understood, although recent studies show that PIRA is associated with increased diffuse neuroaxonal loss, and accelerated brain atrophy, especially in the brain cortex.

On one hand, epidemiology of MS indicates that risk to develop the disease increases to 2-4% in people with first degree affected relative, and to 30-50% in monozygotic twins. Indeed, genomewide association studies evidenced that the haplotype HLA DRB1*1501 is the most significant gene variant that raise the risk of the disease. On the other hand, several environmental risk factors have been identified. It has been reported that in geographic latitude with more temperate climate there is a higher incidence of the disease. This could be due to seasonal exposure to sunlight that influence the synthesis of vitamin D. Other environmental risk factors associated to an enhanced possibility to develop the disease are cigarette smoking, obesity and mononucleosis (3).

Another view regards MS as a primary neurodegenerative disease modified and exacerbated by the following inflammation (7).

 Among the Cap 2 called “Angiogenesis in MS”, the role of angiogenesis in MS is not fully discussed.

According with reviewer’s suggestion we have amended Cap2

In MS, BBB function fails because of a dysregulated paracellular permeability due to the release of histamine, leukotrienes, and chemokines which recruits neutrophils and other cells to invade CNS (12). In particular, in MS, the permeability of the BBB is altered with subsequent transmigration of different subtypes of lymphocytes (Th1 and Th17) and inflammatory cells such as macrophages into the CNS and production of inflammatory mediators (e.g.TNF-α and IFN-γ).

Moreover, it is well known that chronic inflammation and angiogenesis are mutually related and that the immune cells, in particular mast cells, involved in MS pathogenesis are able to secrete cytokines that activate endothelial cells inducing their proliferation and migration (14). Imbalance in angiogenic processes, including excessive angiogenesis, is a hallmark of a number of diseases such as malignant ocular inflammatory disorders, obesity, diabetes, cirrhosis, endometriosis, AIDS and rheumatic diseases (9). In these disorders, the balance between angiogenesis stimulators and inhibitors is impaired, resulting in an “angiogenic switch” (15). Several studies indicate a role of angiogenesis in MS in humans (9, 16-17) as well as in vitro using sera of MS patients in a chick embryo CAM assay (18).

Two hypotheses have been postulated about the role of angiogenesis in MS. In the early phase of the disease, angiogenesis may allow the recruitment of peripheral immune response cells. Bloodborne macrophages infiltrate active multiple sclerosis lesions and remove myelin debris and inflammatory by-products; classically and alternatively activated macrophages, as well as mixed populations, have been described in these lesions (3). On the other hand, in the later phases of MS, angiogenesis could represent an attempt to rescue the hypoperfusion due to the reduction of vascular permeation in CNS (14)

Moreover, the utility of Fig1. as a demonstration of angiogenesis in MS remains unclear and irrelevant (you described a typical MRI finding in a MS patient).

We agreed with the referee and we changed Fig 1.

Cap 3 and 4 are the only that contain relevant information about the topic thereby these paragraphs need to be better developed.

We changed/added the following parts.

In particular, bevacizumab is being tested in patients with neuromyelitis optica, an aggressive inflammatory disease of the central nervous system, characterized by recurrent episodes of optic neuritis and transverse myelitis, which has many symptoms in common with MS (39-42).

Recently, it has been proposed that overactivity of ERK in microglial cells is linked to oligodendrocyte demyelination, highlighting that well-known risk factors for MS, such as hypovitaminosis D, tobacco smoking as well as Epstein Barr Virus infection, are known to downregulate ERK by means of dual specificity phosphatases that normally regulate ERK phosphorylation and activity (61). In conclusion, the MEK/ERK pathway, being involved in many aspects of MS pathogenesis, might be considered a hub in the future development of novel therapeutic strategies against multiple sclerosis.

We also changed Fig. 2 making a new one, more explanatory than the previous one.

In the cap 5, the description of this DMT is not pertinent with the topic “angiogenesis”, moreover table 1 is unnecessary in your manuscript (why you should report data about the administration and side effects of fingolimod, natalizumab, ocrelizumab and others?) , moreover it also contains incorrect data (fingolimod and natalizumab belong to immunomodulators).

According to reviewer’s suggestion we have eliminated table 1.

Moreover, we changed part of the text.

Almost 30 years after the first specific drug for the therapy of MS (beta interferons), numerous drugs are currently available, each with its own characteristics but all with a need for continuous treatment to make inflammation control lasting; some of these (including alemtuzumab and cladribine) cause an immune reconstitution and are administered for short periods in the hypothesis of producing lasting immunological actions (62).

Conclusions are not pertinent with the aims of your paper.

We agree with the reviewer that conclusions should be amended to be pertinents with the aims of the ms, thus we added the following part.

For example, the link between inflammation and angiogenesis, both involved in the pathogenesis of MS, makes compounds endowed with anti-inflammatory as well as anti-angiogenic properties particularly useful. Mounting evidence suggests that natural derivatives could be such compounds, able to exert an efficacious therapeutic effect and some of them are already used in clinical trials.

Finally, the entire manuscript requires an extensive language revision.

We carried out a language revision of the ms.

Reviewer 2 Report

Dear authors 

The manuscript “ANGIOGENESIS AND MULTIPLE SCLEROSIS PATHOGENESIS: a glance at new pharmaceutical approaches.” presents interesting data on the relevance of angiogenesis in multiple sclerosis and some aspects of natural compounds as therapeutic options in MS. However, the connection of those two topics was not achieved ideally in my opinion. In some points, you remain superficial in your conclusions and lack scientific detail. Those flaws in the manuscript have to be addressed prior to publication. I therefore suggest major revisions prior to publication. See in more detail below.

Thank you for your work.

Best regards

Major aspects

·         In the introduction you fail to mention the relatively new concept of PIRA in your thoughts

·         Lines 44-45 your write “MS is not simply a single disease entity but derives from multiple pathogenic pathways…” I’m not sure that this has been proven, I would rephrase this sentence.

·         Figure 1 seems a bit out of context, you do not mention it in your text and the figure itself has no real relation with the text, please adapt.

·         Line 142: “patients with neuromyelitis optica, an aggressive disease which have many symptoms in common with 141 MS.” This description of NMO is insufficient, please be more specific.

·         Lines 172-174: “Thus, the MEK/ERK pathway represents a link that connects all the pathogenic factors involved in MS and ERK could represent a central hub in the future development of novel therapeutic strategies against multiple sclerosis.” I would advise you to tone down this sentence. I guess to claim that this pathway connects all the pathogenic factors involved in MS is an overstatement that you a) cannot prove and b) cannot be sure of, as the pathogenesis of MS is still not completely understood.

·         Lines 201-202: Why do you select only those treatments? There are many other treatments available for MS. Moreover, a high risk of CNS and GI malignancies is not associated with either fingolimod, natalizumab or ocrelizumab.

·         Lines 217-218: “Different herbal medicines are recommended for MS patients.” This sounds as those recommendations would be broadly used in treatment of MS patients, however you cite a single study with often very low n numbers, I would change this phrase.

·         Lines 259-260: “… lack of a therapy that can improve both patient’s quality of life and the outcome of the disease.” This is not entirely true, there are DMTs that change the outcome of the disease and improve the quality of life in patient reported outcomes, please rephrase.

Minor aspects

·         Line 56: you write “cycling ovary”. I’m not sure this is a medical term, please rephrase.

·         Line 69: “BBB function is failed”. Please rephrase to either “has failed” or “fails”.

·         Lines 70-73: “Moreover, in MS, the permeability of the BBB is altered with subsequent transmigration of different subtypes of lymphocytes such as Th1 and Th17 and inflammatory cells such as macrophages into the CNS and production of inflammatory mediators such as TNF-α and IFN-γ…” In this sentence you use the term “such as” three times, please adapt.

·         Line 96: “Experimental Autoimmune Encephalomyelitis (EAE) represents one of the best animal models for MS.” I would advise you to rephrase this to “most commonly used”, as this model has some obvious shortcomings.

·         Lines 125-129: “Moreover, when bevacizumab, a monoclonal anti-VEGF antibody that blocks the VEGF receptor, was administered to EAE mice on the day they began to show clinical symptoms, induced a significant reduction of the number of CD4+ T-cells and inhibits T-cells proliferation in the CNS.” This sentence is not complete, please rephrase and mention, how bevacizumab was administered.

·         Lines 129-130: “However, bevacizumab is also associated with a significantly increased risk of cerebrovascular events, including cerebral hemorrhages and ischemia”. This sentence is only partially true, as only intravenous use of bevacizumab is associated with those risk, not for example intraocular use, please adapt this sentence by adding “intravenous”.

Author Response

Dear Referee,

we have amended the ms according to your comments.

We wish to thank you for the insightful criticisms and comments that helped us to amend and to ameliorate the ms.

REVIEWER 2

Dear authors 

The manuscript “ANGIOGENESIS AND MULTIPLE SCLEROSIS PATHOGENESIS: a glance at new pharmaceutical approaches.” presents interesting data on the relevance of angiogenesis in multiple sclerosis and some aspects of natural compounds as therapeutic options in MS. However, the connection of those two topics was not achieved ideally in my opinion. In some points, you remain superficial in your conclusions and lack scientific detail. Those flaws in the manuscript have to be addressed prior to publication. I therefore suggest major revisions prior to publication. See in more detail below.

Thank you for your work.

Best regards

Major aspects

  • In the introduction you fail to mention the relatively new concept of PIRA in your thoughts

We changed the ms accordingly:

However, some patients can acquire disability also through a relapsing independent progression (PIRA). In particular, PIRA has been shown to occur in typical RRMS populations, challenging the traditional distinction between an early exclusively relapsing phase and a late secondary progressive MS (SPMS). The pathophysiological determinants of PIRA are not understood, although recent studies show that PIRA is associated with increased diffuse neuroaxonal loss, and accelerated brain atrophy, especially in the brain cortex.

  • Lines 44-45 your write “MS is not simply a single disease entity but derives from multiple pathogenic pathways…” I’m not sure that this has been proven, I would rephrase this sentence.

We followed the suggestion of the referee and the phrase is put in a new context with more background concerning the pathogenesis.

It is likely that it is due to genetic predisposition and environmental factors, such as infectious, vascular or metabolic factors, that combine to initiate the disease (3). On one hand, epidemiology of MS indicates that risk to develop the disease increases to 2-4% in people with first degree affected relative, and to 30-50% in monozygotic twins. Indeed, genomewide association studies evidenced that the haplotype HLA DRB1*1501 is the most significant gene variant that raise the risk of the disease. On the other hand, several environmental risk factors have been identified. It has been reported that in geographic latitude with more temperate climate there is a higher incidence of the disease. This could be due to seasonal exposure to sunlight that influence the synthesis of vitamin D. Other environmental risk factors associated to an enhanced possibility to develop the disease are cigarette smoking, obesity and mononucleosis (3). Thus, MS is not simply a single disease entity but could derive from multiple pathogenic factors that lead to final events represented by an inappropriate recognition of an autoantigen on myelinated nerve fibers that recruit macrophages and lymphocytes into the CNS. These events result in white and grey matter demyelination. Another view regards MS as a primary neurodegenerative disease modified and exacerbated by the following inflammation (7).

  • Figure 1 seems a bit out of context, you do not mention it in your text and the figure itself has no real relation with the text, please adapt.

We agree with the referee, thus we changed Fig 1.

  • Line 142: “patients with neuromyelitis optica, an aggressive disease which have many symptoms in common with 141 MS.” This description of NMO is insufficient, please be more specific.

We amended the text according to referee’s suggestion.

We amended the text according to referee’s suggestions.

In particular, bevacizumab is being tested in patients with neuromyelitis optica, an aggressive inflammatory disease of the central nervous system, characterized by recurrent episodes of optic neuritis and transverse myelitis, which has many symptoms in common with MS (39-42).

  • Lines 172-174: “Thus, the MEK/ERK pathway represents a link that connects all the pathogenic factors involved in MS and ERK could represent a central hub in the future development of novel therapeutic strategies against multiple sclerosis.” I would advise you to tone down this sentence. I guess to claim that this pathway connects all the pathogenic factors involved in MS is an overstatement that you a) cannot prove and b) cannot be sure of, as the pathogenesis of MS is still not completely understood.

We agree therefore, following referee’s suggestion we toned down and rephrase the sentence:

In conclusion, the MEK/ERK pathway, being involved in many aspects of MS pathogenesis, is a potential hub in the future development of novel therapeutic strategies against multiple sclerosis.

Moreover we added a new part pointing towards the involvement of ERK in MS.

Recently, it has been showed that overactivity of ERK in microglial cells is linked to oligodendrocyte demyelination, highlighting that well-known risk factors for MS, such as hypovitaminosis D, tobacco smoking as well as Epstein Barr Virus infection, are known to downregulate ERK by means of dual specificity phosphatases that normally regulate ERK phosphorylation and activity (61).

  • Lines 201-202: Why do you select only those treatments? There are many other treatments available for MS. Moreover, a high risk of CNS and GI malignancies is not associated with either fingolimod, natalizumab or ocrelizumab.

We agreed with the referee, thus we have eliminated table 1 and we have also amended the text.

Almost 30 years after the first specific drug for the therapy of MS (beta interferons), numerous drugs are currently available, each with its own characteristics but all with a need for continuous treatment to make inflammation control lasting; some of these (including alemtuzumab and cladribine) cause an immune reconstitution and are administered for short periods in the hypothesis of producing lasting immunological actions (62).

  • Lines 217-218: “Different herbal medicines are recommended for MS patients.” This sounds as those recommendations would be broadly used in treatment of MS patients, however you cite a single study with often very low n numbers, I would change this phrase.

We would like to point out that the cited study is a review containing herbal medicines used for MS.

  • Lines 259-260: “… lack of a therapy that can improve both patient’s quality of life and the outcome of the disease.” This is not entirely true, there are DMTs that change the outcome of the disease and improve the quality of life in patient reported outcomes, please rephrase.

We entirely agree with the referee’s comment. Thus, we changed the text accordingly.

In the case of multiple sclerosis, the need to develop a multi-target drug able to act simultaneously on different pathogenic factors involved, is imperative and could better improve both patient’s quality of life and the outcome of the disease.

Minor aspects

We changed the text according to reviewer’s suggestions.

  • Line 56: you write “cycling ovary”. I’m not sure this is a medical term, please rephrase.

We changed the text

During embryogenesis and after birth, angiogenesis contributes to organ growth but, in adulthood, most blood vessels remain quiescent and angiogenesis occurs only in the ovarian cycle,

  • Line 69: “BBB function is failed”. Please rephrase to either “has failed” or “fails”.

We changed as required.

In MS, BBB function fails because…

  • Lines 70-73: “Moreover, in MS, the permeability of the BBB is altered with subsequent transmigration of different subtypes of lymphocytes such as Th1 and Th17 and inflammatory cells such as macrophages into the CNS and production of inflammatory mediators such as TNF-αand IFN-γ…” In this sentence you use the term “such as” three times, please adapt.

We amended the text.

In particular, in MS, the permeability of the BBB is altered with subsequent transmigration of different subtypes of lymphocytes (Th1 and Th17) and inflammatory cells such as macrophages into the CNS and production of inflammatory mediators (e.g.TNF-α and IFN-γ).

  • Line 96: “Experimental Autoimmune Encephalomyelitis (EAE) represents one of the best animal models for MS.” I would advise you to rephrase this to “most commonly used”, as this model has some obvious shortcomings.

We changed as suggested

Experimental Autoimmune Encephalomyelitis (EAE) represents the most commonly used animal model

  • Lines 125-129: “Moreover, when bevacizumab, a monoclonal anti-VEGF antibody that blocks the VEGF receptor, was administered to EAE mice on the day they began to show clinical symptoms, induced a significant reduction of the number of CD4+ T-cells and inhibits T-cells proliferation in the CNS.” This sentence is not complete, please rephrase and mention, how bevacizumab was administered.

We amended the text accordingly

Moreover, when bevacizumab, a monoclonal anti-VEGF antibody that blocks the VEGF receptor, was administered i.v. to EAE mice on the day they began to show clinical symptoms, a significant reduction of the number of CD4+ T-cells and inhibits T-cells proliferation in the CNS was observed (36).

  • Lines 129-130: “However, bevacizumab is also associated with a significantly increased risk of cerebrovascular events, including cerebral hemorrhages and ischemia”. This sentence is only partially true, as only intravenous use of bevacizumab is associated with those risk, not for example intraocular use, please adapt this sentence by adding “intravenous”.

We changed as suggested by reviewer

However, intravenous administration of bevacizumab is also associated with a significantly increased risk of cerebrovascular events, including cerebral hemorrhages and ischemia (36).

Reviewer 3 Report

This article deals with the pleiotropic roles of VEGF signaling molecule and its dowstream ERK pathway. VEGF mainly induces angiogenesis, which seems to take place in animal models,  but angiogenesis in MS plaques is far from being demonstrated. Other immunomodulatory effects of drugs acting on VEGF/ERK pathway are described. 

Finally authors proposed to examine a list of natural coumpounds which may act on this promising pathway. 

The idea could be interesting but many problems are still unsolved:

* Focal increase of blood perfusion may precede gadolinium enhancement in newly forming plaques. Figure 1 is supposed to illustrate this point, however this figure simply shows a typical enhancing acute plaque centered by a vein. No data are provided about (preceding) perfusion. What does it add to the text?

* Increased perfusion in MS brain is not clearly explained, but it may be the consequence of vasodilatation in response to inflammation, rather than a neoangiogenesis. Acute change in vein diameter within acute plaques, which give conflicting results, is probably related to the interaction between the surrounding edema and the perivascular cuffing of inflammatory cells. 

We are no more aware of changes in the vessel count, length or surface inside plaques or MS brain. Moreover, we were not aware, although data may exist, that neoangiogenesis could play a role in MS plaques or pathophysiology. Indeed Ref 14 points to two articles: Holley et al NeurosciLet 2010 and Ludwin et al J Neuropathol Exp Neurol 2001 (we cannot found the later in pubmed). Article from Holley et al. should be included in the review, since it seems to be the unique article dealing with neoangiogenesis in MS plaques (and not in animal models). 

In our opinion the title of the chapter is rather misleading since data supporting neoangiogenesis in MS (and if confirmed the exact role) are a still uncertain.

* The authors describe that VEGF, via the ERK pathway, is more than an angiogenetic factor and plays a role in T cell response, inflammation, etc. This central role of VEGF/ERK could be a potential future target. As a consequence, the title of the article should be changed since ‘angiogenesis’ does not play a major role in this new pleiotropic target. 

Pathways downstream VEGF ligation are briefly described. The ERK signaling pathway is emphasized. Is there any other significant pathways (e.g. Akt) ?

Table or figure summarizing experimental data would add clarity. Figure 2 is a bit simple in the explanation of such a complex pathway.

* Table 1 provides very general data concerning the main disease modifying drugs in MS, showing that none of them may act on angiogenesis. This table do not add anything clear to the text.  

One would have expected a table summarizing results obtained with drugs acting on angiogenesis in clinical and experimental models.

* Admitting neoangiogenesis as a future therapeutic target, the last chapter reviews the natural compounds which may be used in MS. One would have expected to focus the review on the compounds acting on angiogenesis, whereas many other compounds were proposed (e.g. anti-inflammatory effect of Pulsatilla chinensis) which are out of the scope of the proposed topic. Other examples are highly putative in the setting of MS.

In conclusion, authors proposed a kind of holistic, pleiotropic therapy to simultaneously target multiple pathogenic factors. This is an interesting point of view which should probably be reinforced.

Author Response

Dear Referee,

we have amended the ms according to your comments.

We wish to thank you for the insightful criticisms and comments that helped us to amend and to ameliorate the ms.

REVIEWER 3

This article deals with the pleiotropic roles of VEGF signaling molecule and its dowstream ERK pathway. VEGF mainly induces angiogenesis, which seems to take place in animal models,  but angiogenesis in MS plaques is far from being demonstrated. Other immunomodulatory effects of drugs acting on VEGF/ERK pathway are described. 

Finally authors proposed to examine a list of natural coumpounds which may act on this promising pathway. 

The idea could be interesting but many problems are still unsolved:

* Focal increase of blood perfusion may precede gadolinium enhancement in newly forming plaques. Figure 1 is supposed to illustrate this point, however this figure simply shows a typical enhancing acute plaque centered by a vein. No data are provided about (preceding) perfusion. What does it add to the text?

We agreed with the reviewer, thus we changed Fig.1

* Increased perfusion in MS brain is not clearly explained, but it may be the consequence of vasodilatation in response to inflammation, rather than a neoangiogenesis. Acute change in vein diameter within acute plaques, which give conflicting results, is probably related to the interaction between the surrounding edema and the perivascular cuffing of inflammatory cells.

Following the reviewer’s suggestion we amended the text concerning vasodilatation :

These results indicate an increase in blood vessel density and vasodilation (19).

and the role of inflammatory cells:

Two hypotheses have been postulated about the role of angiogenesis in MS. In the early phase of the disease, angiogenesis may allow the recruitment of peripheral immune response cells. Bloodborne macrophages infiltrate active multiple sclerosis lesions and remove myelin debris and inflammatory by-products; classically and alternatively activated macrophages, as well as mixed populations, have been described in these lesions (3). On the other hand, in the later phases of MS, angiogenesis could represent an attempt to rescue the hypoperfusion due to the reduction of vascular permeation in CNS (14)

We are no more aware of changes in the vessel count, length or surface inside plaques or MS brain. Moreover, we were not aware, although data may exist, that neoangiogenesis could play a role in MS plaques or pathophysiology. Indeed Ref 14 points to two articles: Holley et al NeurosciLet 2010 and Ludwin et al J Neuropathol Exp Neurol 2001 (we cannot found the later in pubmed). Article from Holley et al. should be included in the review, since it seems to be the unique article dealing with neoangiogenesis in MS plaques (and not in animal models).

According to reviewer’s suggestion we added some new references including a recent paper dealing with angiogenesis and MS : Ribatti D, Tamma R, Annese T. (2020) Mast cells and angiogenesis in multiple sclerosis. Inflamm Res.

In our opinion the title of the chapter is rather misleading since data supporting neoangiogenesis in MS (and if confirmed the exact role) are a still uncertain.

As said before we added a recent reference concerning the role of angiogenesis in MS and the involvement of innate cells:  Ribatti D, Tamma R, Annese T. (2020) Mast cells and angiogenesis in multiple sclerosis. Inflamm Res.

* The authors describe that VEGF, via the ERK pathway, is more than an angiogenetic factor and plays a role in T cell response, inflammation, etc. This central role of VEGF/ERK could be a potential future target. As a consequence, the title of the article should be changed since ‘angiogenesis’ does not play a major role in this new pleiotropic target. 

Pathways downstream VEGF ligation are briefly described. The ERK signaling pathway is emphasized. Is there any other significant pathways (e.g. Akt) ?

We agree with the referee that ERK is not the only signal transduction pathway downstream VEGF involved in the pathogenic mechanisms of MS, however, the review of other pathways is behind our aim and could be the topic for a further review.

Table or figure summarizing experimental data would add clarity. Figure 2 is a bit simple in the explanation of such a complex pathway.

 We agree with the referee thus, we changed Fig. 2

* Table 1 provides very general data concerning the main disease modifying drugs in MS, showing that none of them may act on angiogenesis. This table do not add anything clear to the text.  

One would have expected a table summarizing results obtained with drugs acting on angiogenesis in clinical and experimental models.

We agree with the reviewer that Table 1 does not contribute to the clarity of the review, thus we have eliminated it from the revised ms.

* Admitting neoangiogenesis as a future therapeutic target, the last chapter reviews the natural compounds which may be used in MS. One would have expected to focus the review on the compounds acting on angiogenesis, whereas many other compounds were proposed (e.g. anti-inflammatory effect of Pulsatilla chinensis) which are out of the scope of the proposed topic. Other examples are highly putative in the setting of MS.

We understand the point raised by the reviewer, thus we amended this part highlighting that chronic inflammation and angiogenesis are closely linked including in pathogenesis of MS we also provided the reference for this. Moreover, we made the case that natural products are endowed with potent anti-inflammatory effects that, in turn, could play a beneficial role also in preventing pathologic angiogenesis. We hope we have been convincing about that.

It is worth noting that chronic inflammation and neo-angiogenesis can be strictly linked since a number of immune cells such as macrophage, neutrophils, dendritic cells, lymphocytes and mast cells secrete growth factors and cytokines capable to foster angiogenesis from endothelial cells. In particular, it has been recently proposed a role of mast cell in fostering angiogenesis in MS. According to experimental findings mast cells promote angiogenesis both directly by secreting a number of angiogenic factors (i.e. VEGF, FGF-2 etc.) and indirectly by stimulating other inflammatory cells to produce angiogenic molecules (14).

In conclusion, authors proposed a kind of holistic, pleiotropic therapy to simultaneously target multiple pathogenic factors. This is an interesting point of view which should probably be reinforced.

We agreed with the reviewer. Thus, as showed in the previous point, in this revised version, we emphasized the issue that inflammation and angiogenesis, both present in MS, are linked and that natural compounds, being endowed with powerful anti-inflammatory properties, may keep at a bay also pathological angiogenesis. Up to now, few natural compounds have been tested for the effects on angiogenesis other than inflammation. Therefore we would like to focus the attention of the readers on this topic .

Round 2

Reviewer 1 Report

Dear Authors, 

the manuscript is improved. Your work provides a brief and interesting point of view about angiogenesis in pathogenesis and treatment of MS. I think it is not scientifically sound; however, i do not object for publication.

Only a minor further revision is required

-Brief english check

-Delete the sentence: page 2 ln 66 "These events result in white and grey matter demyelination".  I don't know how grey matter could be demyelinated. 

Kind regards

Author Response

REVIEWER 1

Dear Authors, 

the manuscript is improved. Your work provides a brief and interesting point of view about angiogenesis in pathogenesis and treatment of MS. I think it is not scientifically sound; however, i do not object for publication.

Only a minor further revision is required

-Brief english check

-Delete the sentence: page 2 ln 66 "These events result in white and grey matter demyelination".  I don't know how grey matter could be demyelinated. 

Kind regards

We thank the Reviewer for his/her valuable comments that helped us to improve the quality of the ms.

We have deleted the sentence in page 2 line 66 “These events result in white and grey matter demyelination” as suggested by the Reviewer 1, and checked the english grammar.

Reviewer 2 Report

Dear authors 

PIRA is an abbreviation for progression independent of relapse activity, and not relapsing independent progression, please adapt. 

Additionally, you write the following:

Despite this, all these therapies act indirectly on neurodegeneration through the control of neuro-inflammation: in fact, specific treatments to reduce neural damage or repair are not available. 

I would change the phrasing to something like "all these therapies seem to act indirectly...", as a) the mode of action of some MS-therapies is still not completely known and b) neither the pathogenesis of MS.

Otherwise, the manuscript is nicely written.

Thank you for your work.

Best regards

Author Response

REVIEWER 2

Dear authors 

PIRA is an abbreviation for progression independent of relapse activity, and not relapsing independent progression, please adapt. 

Additionally, you write the following:

Despite this, all these therapies act indirectly on neurodegeneration through the control of neuro-inflammation: in fact, specific treatments to reduce neural damage or repair are not available. 

I would change the phrasing to something like "all these therapies seem to act indirectly...", as a) the mode of action of some MS-therapies is still not completely known and b) neither the pathogenesis of MS.

Otherwise, the manuscript is nicely written.

Thank you for your work.

Best regards

We thank the Reviewer for his/her valuable comments that helped us to improve the quality of the ms.

We explained the acronym PIRA (progression independent of relapse activity) in the text and rephrase the sentence in page 6, lines 619-623 “Despite this, all these therapies seem to act indirectly on neurodegeneration through the control of neuro-inflammation. However, the mechanisms of action of some available therapies and the pathogenesis of MS are still incompletely known, thus specific treatments to reduce neural damage or repair are not yet available”

Reviewer 3 Report

This articles deals with a three step syllogism: neoangiogenesis is observed in MS; VEGF/MAPK/ERK1/2 pathway is involved in angiogenesis; which natural drug may target this pathway. 

Our major reserve comes from the very uncertain neoangiogenesis in MS lesions. This point should examined deeply in details before asking how it may be involved in the disease, or even targeted. 

Data concerning neoangiogenesis were very short (p3 L108-114) and far being convincing.

Minor concerns:

Figure was supposed to illustrate ref19 assertion: focal increase of perfusion weeks before enhancement, however the lesion depicted is enhancing.

Although figure 1 was modified, it still remains problematic. Figure is a low quality screen capture.

Two profile curves are shown: where are the voxel of interest?  Is it really sure that these curves are abnormal (we are not expert in MRI, but these curves look like normal profiles). 

Reader would expect comparison of normal and pathological profile. Moreover, angiogenesis is not an expected feature of common MS plaques, therefore this assertion of hyper-perfusion should be strongly supported by bibliography, and reader would expect from the figure to contrast one picture of a normally expected perfusion of an MS plaques, and another picture of a MS plaque with neoangiogenesis.

High CBV in acute MS lesions is unexpected (see Li, 2014 J Neurol Neurophysiol PMC4309012).

Figures are not linked with text.

Author Response

REVIEWER 3

This articles deals with a three step syllogism: neoangiogenesis is observed in MS; VEGF/MAPK/ERK1/2 pathway is involved in angiogenesis; which natural drug may target this pathway. 

Our major reserve comes from the very uncertain neoangiogenesis in MS lesions. This point should examined deeply in details before asking how it may be involved in the disease, or even targeted. 

Data concerning neoangiogenesis were very short (p3 L108-114) and far being convincing.

We thank the Reviewer for his/her valuable comments that helped us to improve the quality of the ms.

We agree with the Reviewer for his/her concerns about the presence of angiogenesis in MS lesions. On this issue, we added some references supporting the hypothesis that increase in cerebral blood volume and cerebral blood flow, observed in MS lesions is associated with neoangiogenesis. In particular, we reported the results of a MRI study  at page 3, line 163-176: “These results indicate an increase in blood vessel density and vasodilation. In particular, Bester et al. reported a study in which 33 patients with relapsing-remitting MS underwent four monthly 3T MRI scans including dynamic susceptibility contrast perfusion-weighted MRI. They show that high inflammatory lesions are associated with increased cerebral blood volume (CBV) and cerebral blood flow (CBF) (19). CBV has been proven to be a sensitive marker of angiogenesis, with significant correlations between the increase of CBV and CBF values and microvessel density [20]. Notably, the arrangement of angiogenic vessels in MS appears orderly whereas tumor angiogenesis is chaotic [21] suggesting a reparative response to injury [22]. Although no immunohistochemical data is currently available, it is possible that the elevated CBV/CBF values in the normal appearing white matter may be indicative of angiogenesis occurring additionally to inflammation-induced vasodilatation [23]”.

Minor concerns:

Figure was supposed to illustrate ref19 assertion: focal increase of perfusion weeks before enhancement, however the lesion depicted is enhancing.

Although figure 1 was modified, it still remains problematic. Figure is a low quality screen capture.

Two profile curves are shown: where are the voxel of interest?  Is it really sure that these curves are abnormal (we are not expert in MRI, but these curves look like normal profiles). 

Reader would expect comparison of normal and pathological profile. Moreover, angiogenesis is not an expected feature of common MS plaques, therefore this assertion of hyper-perfusion should be strongly supported by bibliography, and reader would expect from the figure to contrast one picture of a normally expected perfusion of an MS plaques, and another picture of a MS plaque with neoangiogenesis.

High CBV in acute MS lesions is unexpected (see Li, 2014 J Neurol Neurophysiol PMC4309012).

 Following the reviewer’s comment we decide to eliminate Figure 1
